# Low expression of EXOSC2 protects against clinical COVID-19 and impedes SARS-CoV-2 replication

Tobias Moll[1], Valerie Odon[2], Calum Harvey[1], Mark O Collins[3], Andrew Peden[3], John Franklin[1], Emily Graves[1], Jack NG Marshall[1], Cleide dos Santos Souza[1], Sai Zhang[4,5], Lydia Castelli[1], Guillaume Hautbergue[1], Mimoun Azzouz[1], David Gordon[6,7], Nevan Krogan[6,7,8,9], Laura Ferraiuolo[1], Michael P Snyder[4,5], Pamela J Shaw[1], Jan Rehwinkel[2], Johnathan Cooper-Knock[1]

**New therapeutic targets are a valuable resource for treatment of SARS-CoV-2 viral infection. Genome-wide association studies have identified risk loci associated with COVID-19, but many loci are associated with comorbidities and are not specific to host–virus interactions. Here, we identify and experimentally validate a link between reduced expression of EXOSC2 and reduced SARS-CoV-2 replication. EXOSC2 was one of the 332 host proteins examined, all of which interact directly with SARS-CoV-2 proteins. Aggregating COVID-19 genome-wide association studies statistics for gene-specific eQTLs revealed an association between increased expression of _EXOSC2_ and higher risk of clinical COVID-19. EXOSC2 interacts with Nsp8 which forms part of the viral RNA polymerase. EXOSC2 is a component of the RNA exosome, and here, LC-MS/MS analysis of protein pulldowns demonstrated interaction between the SARS-CoV-2 RNA polymerase and most of the human RNA exosome components. CRISPR/Cas9 introduction of nonsense mutations within _EXOSC2_ in Calu-3 cells reduced EXOSC2 protein expression and impeded SARS-CoV-2 replication without impacting cellular viability. Targeted depletion of EXOSC2 may be a safe and effective strategy to protect against clinical COVID-19.**

## Introduction

Infection with severe acute respiratory syndrome coronavirus 2 (SARS-CoV-2) giving rise to COVID-19 has caused a global pandemic with almost unprecedented morbidity and mortality (Dong et al, 2020). Vaccination efforts have led to early successes (Shilo et al, 2021), but the prospect of evolving variants capable of immune escape (Darby & Hiscox, 2021) and a time limit on vaccine effectiveness (Pouwels et al, 2021) highlight the importance of efforts to better understand COVID-19 pathogenesis and to develop effective treatments.

SARS-CoV-2 gains entry to host cells in the upper airway via interaction with the cell-surface proteins ACE2 and TMPRSS2 (Hoffmann et al, 2020), but like all viruses, the process of viral replication requires interaction with a range of host proteins intracellularly. Early work to determine important interactions between viral and host proteins outlined a set of 332 high-confidence interactions (Gordon et al, 2020). We hypothesised that variation in function and expression of host proteins involved in these interactions could modify SARS-CoV-2 replication and potentially increase or decrease the risk of symptomatic infection.

To date, therapeutic approaches to COVID-19 have focused on modulation of the host immune response. Severe infection is thought to result from a combination of uncontrolled viral replication and a late hyperinflammatory response leading to acute respiratory distress syndrome (Brodin, 2021). As a result, current therapeutic approaches seek to either boost host immunity (e.g., vaccines) (Olliaro et al, 2021; Zhang et al, 2022), reduce viral replication (e.g., molnupiravir [Jayk Bernal et al, 2021]), or reduce hyperinflammation (e.g., dexamethasone [RECOVERY Collaborative Group et al, 2021]). Very few strategies have sought to modify a host protein which interacts with the virus: an exception is camostat, a TMPRSS2 inhibitor, for which there is no good evidence of effectiveness (Gunst et al, 2021).

We performed an unbiased genetic screen using eQTL data from GTEx (Lonsdale et al, 2013), describing gene expression changes in lung tissue. We tested all 332 genes encoding proteins which interact with SARS-CoV-2 proteins, to determine whether gene expression was linked to the risk of clinically symptomatic COVID-19. Increased EXOSC2 expression within the lung was significantly associated with a higher risk of COVID-19 after stringent Bonferroni multiple testing correction. _EXOSC2_ encodes a component of the

[1]Sheffield Institute for Translational Neuroscience, University of Sheffield, Sheffield, UK   [2]Medical Research Council Human Immunology Unit, Medical Research Council Weatherall Institute of Molecular Medicine, Radcliffe Department of Medicine, University of Oxford, Oxford, UK   [3]School of Biosciences, University of Sheffield, Sheffield, UK   [4]Department of Genetics, Stanford University School of Medicine, Stanford, CA, USA   [5]Center for Genomics and Personalized Medicine, Stanford University School of Medicine, Stanford, CA, USA   [6]Department of Cellular and Molecular Pharmacology, University of California San Francisco, San Francisco, CA, USA   [7]Quantitative Biosciences Institute, University of California San Francisco, San Francisco, CA, USA   [8]Gladstone Institute of Data Science and Biotechnology, J. David Gladstone Institutes, San Francisco, CA, USA   [9]Quantitative Biosciences Institute (QBI) COVID-19 Research Group (QCRG), San Francisco, CA, USA

Correspondence: j.cooper-knock@sheffield.ac.uk

RNA exosome. We engineered Calu-3 cells to reduce expression of EXOSC2 and demonstrated a significant suppression of SARS-CoV-2 replication. Transcriptome analysis revealed that reduced expression of EXOSC2 leads to an up-regulation of *OAS* gene expression which is independent of infection or inflammation, possibly as part of a homoeostatic response (Mullani et al, 2021). OAS proteins are key mediators of viral RNA degradation (Choi et al, 2015) and have been linked to a successful immune response against SARS-CoV-2 (Wickenhagen et al, 2021; Huffman et al, 2022); it is likely that OAS protein up-regulation is one reason for the negative effect of reduced EXOSC2 on SARS-CoV-2 replication. Our data suggest that EXOSC2 is a novel therapeutic target for preventing uncontrolled replication of SARS-CoV-2 after infection. Our approach is summarised in Fig 1A.

# Results

## Unbiased genetic screen highlights RNA exosome components in the defence against SARS-CoV-2

We hypothesised that changes in expression of host proteins which interact with viral proteins could modify the risk of clinical COVID-19.

To test this, we focused on genetic variation associated with gene expression changes within lung tissue available from GTEx (v7) (Lonsdale et al, 2013). We measured risk of clinical COVID-19 using a specific set of symptoms (Menni et al, 2020) rather than a positive test to maximise detection of clinically significant COVID-19. Risk of clinical COVID-19 was assigned to specific genetic variants by genome-wide association studies (GWAS) (COVID-19 Host Genetics Initiative, 2021). We did not focus on hospitalised or severe COVID-19 to minimise confounding by host comorbidities and immune function, which have been closely associated with COVID-19 mortality (Brodin, 2021; Fathi et al, 2021) but may not reflect intracellular interactions between host and viral proteins.

Lung eQTLs were available for 208 of 332 high-confidence COVID-19 interacting partners (Gordon et al, 2020). Using this information, we aggregated genetic variants according to their effect on expression of COVID-19 interacting partners. We then used GWAS data to test whether expression changes are associated with higher or lower risk of clinical COVID-19 (see the Materials and Methods section). After Bonferroni multiple testing correction, only EXOSC2 expression was significantly associated with COVID-19 risk (Table S1 and Fig 1B); higher expression of EXOSC2 was associated with higher risk

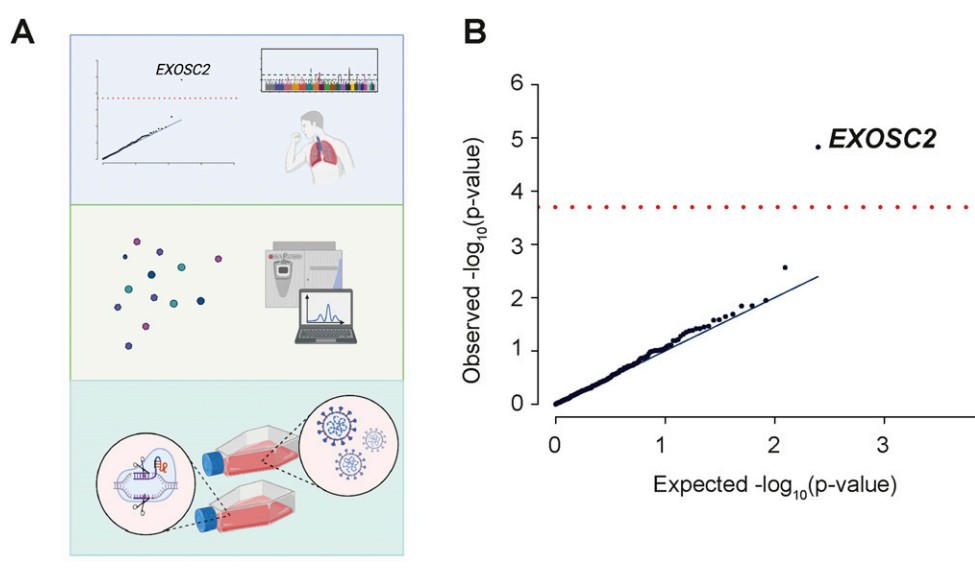

**Figure 1. Unbiased screen of host proteins identified as high-confidence interacting partners of SARS-CoV-2 proteins links RNA exosome components to risk of clinical COVID-19.**
**(A)** Schematic of the study design. Known host-viral interactions were screened for disease-association by combining lung-specific eQTLs with a genome-wide association studies for COVID-19 symptoms. Identification of a positive correlation between EXOSC2 expression and increased severity of COVID-19 led to further study of interactions between the SARS-CoV-2 polymerase and the entire human RNA exosome by AP-MS. Finally, CRISPR editing of *EXOSC2* within human lung cells and subsequent infection with SARS-CoV-2 facilitated validation of the relationship between EXOSC2 expression and viral replication and interrogation of the underlying biological mechanism. **(B)** Lung eQTLs were used to group genetic variants according to their effect on expression of 332 host genes encoding proteins which interact with viral proteins. Only expression of *EXOSC2* was significantly associated with clinical risk of COVID-19 after Bonferroni multiple testing (red line). **(C, D)** Lung eQTLs were used to group genetic variants according to their effect on expression of all genes encoding components of the RNA exosome. Expression levels of EXOSC7, EXOSC9, and EXOSC2 were significantly linked to clinical COVID-19, and in each case, higher expression was associated with higher risk of infection. *P* = 0.05 is indicated by a red dashed line.

of clinical COVID-19 ($Z$ = +4.32, $P$ = 1.5 × 10$^{-5}$). There was no evidence of statistical inflation ($\lambda_{GC}1000$ = 0.99).

### Affinity purification confirms interaction of the SARS-CoV-2 viral polymerase with the host RNA exosome

EXOSC2 is bound by Nsp8 (Gordon et al, 2020) which forms part of the SARS-CoV-2 polymerase. We hypothesised that the SARS-CoV-2 RNA polymerase may interact with the entire host RNA exosome complex. The original study of high-confidence interactions between viral proteins and host proteins necessarily included stringent thresholds and may have missed certain interactions. Moreover, the viral polymerase includes Nsp7 in addition to Nsp8 (Hillen et al, 2020). To explore this hypothesis, we performed pulldown experiments using Strep-tagged Nsp8 co-expressed with untagged Nsp7. LC-MS/MS analysis of replicates Strep-Nsp8 and control pulldowns were performed. Statistical analysis of label-free quantification data confirmed the presence of EXOSC2, EXOSC3, EXOSC5, and EXOSC8 in nsp8 pulldowns (Gordon et al, 2020) and identified additional components of the RNA exosome (EXOSC1, EXOSC4, EXOSC5, EXOSC6, EXOSC7, EXOSC8, EXOSC9, and EXOSC10) (FDR < 0.05, permutation test, Fig 2 and Table S2, see the Materials and Methods section).

We wondered whether the interaction with RNA exosome components was specific to SARS-CoV-2. We used data from a previous affinity purification study of interactions between components of both SARS-CoV-1 and SARS-CoV-2 within A549 cells engineered to express ACE2 to facilitate viral entry (Stukalov et al, 2021). No component of the host RNA exosome was discovered to interact with SARS-CoV-1 including EXOSC2. This suggests that the interactions we observe may be specific to SARS-CoV-2. This same work compared interactions of SARS-CoV-1 and SARS-CoV-2 and

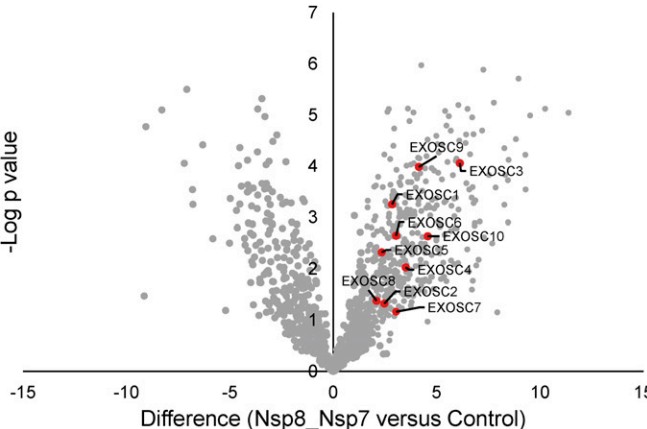

**Figure 2. AP-MS analysis confirms the interaction of the SARS-CoV-2 RNA polymerase with EXOSC2 and most of the components of the host RNA exosome.**
Replicate affinity purifications of HEK293T cells expressing Strep-Nsp8 and untagged Nsp7 and control purifications (mock-transfected) were analysed by label-free quantitative mass spectrometry. Volcano plot of Strep-Nsp8 pulldowns from cells co-expressing Nsp7 compared with mock-transfected cells. RNA exosome complex proteins within the set of enriched proteins are labelled.

determined that 26% of its host interactions are specific to SARS-CoV-2.

### Expression of additional RNA exosome components are linked to the defence against SARS-CoV-2

Our immunoprecipitation data strongly suggest that the SARS-CoV-2 polymerase interacts with the entire host RNA exosome complex. In view of this, we analysed the association between expression of other RNA exosome components with risk for clinical COVID-19. Lung eQTLs were available for EXOSC2, EXOSC3, EXOSC5, EXOSC6, EXOSC7, EXOSC9, EXOSC10, and DIS3. Like EXOSC2, higher expression levels of EXOSC7 and EXOSC9 are significantly associated with higher risk for clinical COVID-19 ($P$ < 0.05, Fig 1C and D).

### Infectivity assays reveal reduced viral replication with reduced EXOSC2 expression

We discovered that higher expression of RNA exosome components in lung tissue is associated with higher risk of clinical COVID-19. Next, we sought to provide experimental support for our observations and explore an underlying mechanism. We used CRISPR/Cas9 to introduce loss-of-function mutations within EXOSC2 in Calu-3 cells. Calu-3 cells are a lung cancer cell line capable of supporting robust SARS-CoV-2 entry and replication (Chu et al, 2020) and a recommended model for viral infection of nasal and bronchotracheal epithelium (Cagno, 2020). Calu-3 cells grow in tight monolayers, present villi, and are capable of secreting mucins.

We designed an sgRNA to target exon 1 of EXOSC2, so as to introduce a series of indels by CRISPR/SpCas9 editing (see the Materials and Methods section). The efficient introduction of nonsense mutations within edited cells was confirmed by Sanger sequencing and waveform decomposition analysis (Fig S1A and B) (Conant et al, 2022). We estimate that ~60% of alleles were successfully edited (Fig S1B); a polyclonal population serves to increase confidence in our downstream analysis. We successfully achieved efficient reduction of EXOSC2 mRNA levels (63% reduction in mRNA expression, Fig S1C, see the Materials and Methods section) and protein levels (Fig S1D, see the Materials and Methods section). The RNA exosome contributes to several RNA processes in the cells, for example, it is critical for the production of mature rRNA (Allmang et al, 2000; Klinge & Woolford, 2019). Reduced expression of EXOSC2 in this cell type was not associated with detectable cell death (MTT assay, Fig 3A, see the Materials and Methods section), suggesting that a reduction in RNA exosome expression may be well tolerated in human lung cells. We used unedited wild-type (WT) Calu-3 cells and a commercially available control sgRNA targeting HPRT as negative controls.

CRISPR-edited Calu-3 cells and control cells were infected with the SARS CoV-2 strain Victoria using a MOI of 1. We confirmed viral replication by absolute RT-qPCR quantification of viral genomes and by the median tissue culture infectious dose (TCID50) (see the Materials and Methods section). As a positive control, we used a neutralising antibody for SARS-COV-2 (see the Materials and Methods section). To achieve absolute quantification of viral genomes, we used two nucleocapsid gene targets, N1 and N2 (Holshue et al, 2020) (see the Materials and Methods section). As predicted,

**Figure 3. Reduced expression of EXOSC2 in Calu-3 cells is not toxic and leads to reduced viral replication.**
**(A)** Calu-3 cells were targeted with the indicated sgRNAs and cell viability was analysed by MTT assay. Data for unedited control cells were set to 100%. **(B, C, D, E, F)** Calu-3 cells targeted with sgRNAs and subsequently reconstituted with EXOSC2 as indicated were infected with SARS-CoV-2 (MOI = 1) for 17 h. As a negative control, cells infected with virus were exposed to a neutralising antibody. **(B)** Viral titres in supernatant samples were analysed by TCID50 assay. **(C, D)** Viral RNA levels were measured by absolute RT-qPCR quantification of N1 and N2 SARS-CoV-2 genomic RNA. **(E)** Viral genomic reads as a proportion of total RNA sequencing reads. **(F)** Viral genomic RNA sequencing reads mapped across the SARS-CoV-2 genome by normalised read-depth; colours represent distinct viral transcripts. Data are from three independent biological repeats. In panels (A, B, C, D, E), individual data points are shown with mean and SE. Significance was tested by the paired $t$ test, and $P$-values are indicated.

reduced expression of EXOSC2 was associated with a significant reduction in viral infectivity (72% reduction, $P$ = 0.004, paired $t$ test, Fig 3B) and in viral genome replication (N1: 62% reduction, $P$ = 0.02; N2: 74% reduction, $P$ = 0.03; paired $t$ test, Fig 3C and D) compared with infection of WT unedited Calu-3 cells.

As a positive control, we performed reconstitution of EXOSC2 by overexpression of an sgRNA-resistant plasmid encoding EXOSC2 (see the Materials and Methods section). Reconstitution was confirmed by immunoblotting (Fig S1D). Reconstitution of EXOSC2 followed by infection with SARS-CoV-2 led to increased viral infectivity (93% increase, Fig 3B) and replication (N1: 44% increase; N2: 32% increase; Fig 3C and D), although these changes were not statistically significant. The failure of complete recovery of viral infectivity and replication may be because both EXOSC2 depletion and reconstitution were variable across the population of cells and/ or because reduced expression of EXOSC2 changes cellular homoeostasis in such a way that is not easily reversible. One possibility resulting from our CRISPR methodology is that truncated EXOSC2 protein is produced from the edited locus that is both undetectable with the antibody we used and which reduces function of endogenous EXOSC2 via a dominant negative effect. Another possibility is that depletion of EXOSC2 results in destabilisation of other RNA exosome components. To address this possibility, we performed immunoblotting of the total set of RNA exosome components in the presence and absence of EXOSC2 editing. However, despite three biological repeats, no other exosome component was significantly reduced in the context of EXOSC2 depletion (Fig S2).

### Transcriptome analysis links reduced EXOSC2 function to reduced SARS-CoV-2 replication and increased expression of OAS proteins

Infectivity assays confirmed our hypothesis based on genetic analyses that reduced EXOSC2 function is protective against SARS-CoV-2 infection. To gain further insight into the biological mechanism underpinning our observations, we analysed changes in the total transcriptome of Calu-3 cells in the presence/absence of SARS-CoV-2 infection, both with and without CRISPR editing of *EXOSC2*. We performed RNA sequencing using three biological repeats for each condition (see the Materials and Methods section).

Mapping of RNA sequencing reads to the SARS-CoV-2 genome (see the Materials and Methods section) confirmed our previous finding that there are significantly fewer viral reads in the presence of reduced EXOSC2 expression (Fig 3E). To determine whether this was because of differential expression of viral subgenomic RNAs, we examined read-depth across the SARS-CoV-2 genome; there was no significant difference in the presence/absence of reduced EXOSC2 expression (Fig 3F). We compared the ratio of reads mapped to Orf1 compared with Orf10 as a measure of transcription of subgenomic RNA (see the Materials and Methods section), but there was no significant difference between the sample groups ($t$ test, $P$ = 0.14). Overall, we conclude that reduced expression of EXOSC2 impacts overall viral replication rather than altering expression of specific viral transcripts.

Next, we analysed RNA sequencing reads mapped to the human genome (see the Materials and Methods section). Principal components analysis revealed good consistency between biological replicates (Fig 4A). As expected, the first principal component and

the largest change in gene expression were associated with the presence or absence of SARS-CoV-2 infection. The most significant functional enrichment by adjusted $P$-value, within the set of 2,445 genes up-regulated in WT Calu-3 cells after SARS-CoV-2 infection (Fig S3), was the KEGG pathway "TNF signalling pathway" (Fisher exact test, adjusted $P$ = $2.28 \times 10^{-21}$, OR = 7.87), suggesting that this represents the immune response to acute SARS-CoV-2 infection. Our observations are consistent with previous literature: the 2,445 genes were highly enriched with reported gene expression changes in Calu-3 cells infected with SARS-CoV-2 (Fisher exact test, adjusted $P$ = $9.44 \times 10^{-245}$, OR = 27.78) (Wyler et al, 2021). We examined expression of these 2,445 genes across all conditions and concluded that the overall cellular response to SARS-CoV-2 infection was independent of *EXOSC2* gene editing (Figs 4B and S3). Indeed, *IL6*, a key inflammatory gene, which is up-regulated in patients suffering COVID-19 (Manjili et al, 2020), was up-regulated in the context of SARS-CoV-2 infection ($P$-value = $6.44 \times 10^{-73}$, FC = 3.32) but down-regulated in uninfected EXOSC2 edited Calu-3 cells compared with unedited cells ($P$ = $3.45 \times 10^{-10}$, FC = 0.67, Fig 4C). Reduced EXOSC2 expression within infected Calu-3 cells produced 903 differentially expressed genes (Fig S3) which represents the effects of EXOSC2 depletion combined with reduced viral replication (Fig 3E).

Next, we examined changes in gene expression in Calu-3 cells with reduced EXOSC2 expression before SARS-CoV-2 infection. In uninfected cells, there were 364 differentially expressed genes as a result of EXOSC2 depletion (Fig 4C). This set of genes was not enriched with "TNF signalling pathway" genes ($P$ = 0.68), unlike the response to SARS-CoV-2 infection. Unexpectedly, the most significant functional enrichment by adjusted $P$-value within this gene set was the KEGG pathway "Coronavirus disease" (Fisher exact test, adjusted $P$-value = $7.7 \times 10^{-4}$, OR = 4.64) which included up-regulation of *OAS1* (genewise exact test [Robinson & Smyth, 2008], $P$ = $4.68 \times 10^{-8}$, FC = 1.21, Fig 4E) and *OAS3* ($P$ = $3.81 \times 10^{-7}$, FC = 1.20, Fig 4F). Both *OAS1* and *OAS3* genes encode enzymes which activate ribonuclease L to degrade intracellular double-stranded RNA as part of the antiviral response (Choi et al, 2015) and cellular homoeostasis (Mullani et al, 2021).

*OAS* genes form part of the set of interferon-stimulated genes (ISGs) (Schoggins et al, 2011). We wondered whether EXOSC2 knockdown produced an indirect up-regulation of ISGs, perhaps via increased concentration of dsRNAs. Of 397 ISGs, only 14 genes were differentially expressed in uninfected Calu-3 cells after CRISPR editing of EXOSC2 which is less than would be expected by chance at a 5% significance level. A heatmap of ISGs across all four conditions (Fig 4D) demonstrates that ISGs are up-regulated after SARS-CoV-2 infection but do not efficiently separate cell lines based on EXOSC2 editing status. Overall, we conclude that the observed up-regulation of OAS genes in the context of reduced EXOSC2 expression is specific and not associated with activation of all ISGs.

## Discussion

The global COVID-19 pandemic entered a new stage when the long-term effectiveness of vaccines was first questioned (Pouwels et al, 2021). New therapeutic strategies are required to prevent COVID-19 infection and associated morbidity and mortality. Our study is data-driven; we have harnessed the power of large-scale GWAS based on

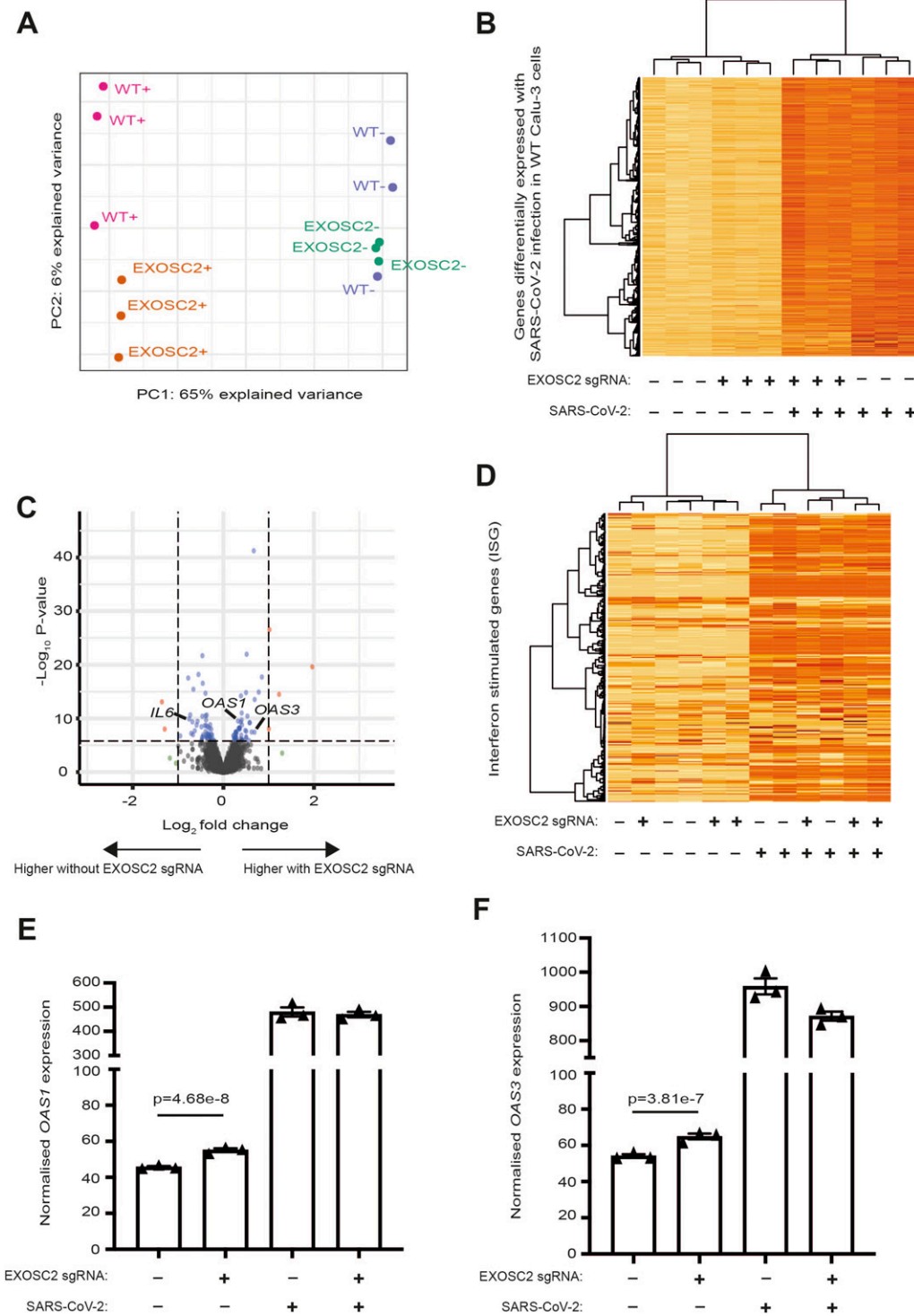

**Figure 4. Transcriptomic analysis confirmed the inflammatory response to SARS-CoV-2 infection of Calu-3 cells and identified up-regulation of *OAS* genes in the context of reduced EXOSC2 expression.**

RNA for sequencing was extracted from Calu-3 cells in the presence and absence of CRISPR editing with sgRNA targeted against EXOSC2; with and without infection with SARS-CoV-2 (MOI = 1) at 17 h; three biological replicates were obtained for all conditions. **(A)** First and second principal components for total gene expression across all sequenced samples. Samples include WT unedited Calu-3 cells and EXOSC2 edited Calu-3 cells; ± indicates the presence/absence of SARS-CoV-2 infection. **(B)** Heatmap representation of genes up-regulated in WT cells in the presence of SARS-CoV-2 infection. A darker colour indicates higher expression. **(C)** Volcano plot to compare gene expression in uninfected Calu-3 cells with and without CRISPR editing of EXOSC2. Dotted lines represent fold change of ± 2 and a Bonferroni multiple testing threshold for *P*-value by the genewise exact test. **(D)** Heatmap representation of 397 interferon-stimulated genes (Schoggins et al, 2011) across all sequenced samples. **(D, E, F)** Normalised expression of OAS1 (D) and OAS3 (E) in all four conditions.

real-world observation to identify factors which reduce the risk for clinical COVID-19 in human patients. However, we have also taken advantage of insights at a molecular level to identify candidate host–virus interactions which may influence SARS-CoV-2 replication (Gordon et al, 2020). By validating our work using live virus, we have demonstrated the validity of our findings to reduce viral replication.

The RNA exosome functions in RNA quality control; aberrant or unwanted RNAs are captured and degraded by the complex (Kilchert et al, 2016). EXOSC2 and EXOSC3 form part of a cap structure with RNA-binding activity responsible for passing substrate RNAs into a barrel-like structure formed by components including EXOSC5 and EXOSC8, from which RNAs then access the catalytically active core. RNA degradation is performed by EXOSC10, DIS3, or DIS3L; the precise catalytic subunit varies by the subcellular location (Tomecki et al, 2010). The RNA exosome has been implicated in the antiviral response (Molleston & Cherry, 2017) and has been observed to associate with other RNases.

A key question is the mechanism underlying the protection we have observed in cells and patients with reduced EXOSC2 expression. Although we have not conclusively answered this question, we highlight a number of possibilities. Previously, SKIV2L, a component of the RNA exosome-activating SKI complex, was shown to limit baseline type I IFN responses, which are induced by RNA sensors in settings of *SKIV2L* deficiency (Eckard et al, 2014). Moreover, pharmacological inhibition of the SKI complex limits SARS-CoV-2 replication (Weston et al, 2020). We did not observe broad ISG induction in EXOSC2-depleted cells, suggesting that reduced SARS-CoV-2 replication was not simply a consequence of elevated baseline ISG expression in these cells.

We identified EXOSC2 because it interacts with the SARS-CoV-2 polymerase (Gordon et al, 2020). Interaction between the host RNA exosome and the viral RNA polymerase is important for viral replication for influenza A virus and Lassa virus (Ho et al, 2021). These viruses do not encode their own capping enzymes and so take advantage of host caps which are a byproduct of RNA degradation by the RNA exosome. Loss of RNA exosome function has been previously shown to protect against influenza A virus infection (Rialdi et al, 2017). However, current knowledge regarding coronaviruses and SARS-CoV-2 in particular indicates that this virus encodes its own mRNA capping machinery (Viswanathan et al, 2020), suggesting that this is not the mechanism underpinning our observations. Moreover, we did not observe a difference in production of specific viral transcripts in the presence of reduced EXOSC2 expression, contrary to what might be expected with failure of viral mRNA capping.

Finally, the RNA exosome functions in degradation of exogenous and endogenous RNAs alongside OAS proteins; indeed, reciprocal up-regulation of OAS proteins has been observed in the context of EXOSC3 depletion (Mullani et al, 2021). The physical association between SARS-CoV-2 and the RNA exosome suggests that the virus is relatively protected against degradation by the exosome, but its vulnerability to OAS proteins is well described (Wickenhagen et al, 2021; Huffman et al, 2022). However, although we observed a modest up-regulation of *OAS* transcripts in the context of reduced EXOSC2 expression in the absence of SARS-CoV-2 infection, it is not possible to assign a functional significance to these changes based on our current data. Future work should examine whether manipulation of OAS proteins modulates the effect of EXOSC2 on viral replication.

Up-regulation of the p46 splice variant of *OAS1* has been specifically associated with protection against severe COVID-19 (Huffman et al, 2022). Production of the p46 variant is contingent upon a G allele for SNP rs10774671, whereas Calu-3 cells are homozygous A (Iida et al, 2021 *Preprint*) and therefore cannot express the p46 variant. It follows that if the reduced viral replication we observed in Calu-3 cells expressing low levels of EXOSC2 is mediated via OAS protein up-regulation, then it must be achieved only through OAS3 and not OAS1. We conclude that our study of Calu-3 cells may therefore underestimate the effect of OAS protein up-regulation, in comparison with our genetic study which focused on a population where OAS1 p46 expression will be frequent.

Our genetic study suggests that reduced expression of EXOSC2 is well tolerated in a significant proportion of the population who are relatively immune to clinical COVID-19. Indeed, we note that our genetically edited cells did not show excess toxicity. Future structural biology work will be necessary to determine the mechanism linking the interaction between SARS-CoV-2 and the host RNA exosome and changes in viral replication; however, we have suggested candidate mechanisms based on our analyses. We have identified a new therapeutic target with the potential to protect against COVID-19. We anticipate that our work will lead to new understanding and new therapies.

# Materials and Methods

### Association of lung gene expression with risk of COVID-19

Association testing was performed using FUSION (Gusev et al, 2016) using pre-computed weights for lung gene expression from GTEx (v7) (Lonsdale et al, 2013). All eQTLs were used regardless of significance. We used the ANA7 phenotype from DF2 which included 1,294 individuals with clinical evidence of COVID-19 and 26,969 asymptomatic controls.

### Affinity purification of Nsp8-associated complexes and mass spectrometry analysis

HEK293T cells were transfected with Strep-Nsp8 (Gordon et al, 2020) and Strep-Nsp8 together with untagged Nsp7 using PEI for 24 h. Mock-transfected cells were used for control purifications. Cell pellets were collected, proteins lysates were prepared, and affinity purification was performed using four biological replicates of each group as previously described (Gordon et al, 2020). Elution of affinity-purified proteins was performed by incubation of MagStrep "type 3" beads with 50 $\mu$l of elution buffer (5% SDS, 50 mM Tris, pH 7.4) at 70°C for 15 min. Protein reduction was performed by adding TCEP to a final concentration of 5 mM and incubation at 70°C for 15 min and alkylation by adding iodoacetamide to a final concentration of 10 mM and incubation at 37°C for 30 min. Sample clean-up was performed using suspension trapping (S-Trap) according to the manufacturer's instructions (ProtiFi). Tryptic digestion was performed by adding 1 $\mu$g of trypsin (sequencing grade; Pierce) and incubating at 47°C for 60 min. Eluted peptides were dried to completion in a vacuum concentrator (Eppendorf).

Samples were re-suspended in 40 $\mu$l of 0.5% formic acid, and 18 $\mu$l was analysed by nanoflow LC-MS/MS using an Orbitrap Elite (Thermo Fisher Scientific) hybrid mass spectrometer equipped with a nanospray source, coupled to an Ultimate RSLCnano LC System (Dionex) at the biOMICS facility at the University of Sheffield. The system was controlled by Xcalibur 3.0.63 (Thermo Fisher Scientific) and DCMSLink (Dionex). Peptides were desalted on-line using an Acclaim PepMap 100 C18 nano/capillary BioLC, 100A nanoViper 20 mm × 75 $\mu$m I.D., and particle size 3 $\mu$m (Thermo Fisher Scientific) at a flow rate of 5 $\mu$l/min and then separated using a 125-min gradient from 5% to 35% buffer B (0.5% formic acid in 80% acetonitrile) on an EASY-Spray column, 50 cm × 50 $\mu$m ID, PepMap C18, 2 $\mu$m particles, and 100 Å pore size (Thermo Fisher Scientific) at a flow rate of 0.25 $\mu$l/min. The Orbitrap Elite was operated with a cycle of one MS (in the Orbitrap) acquired at a resolution of 60,000 at m/z 400, with the top 20 most abundant multiply charged (2+ and higher) ions in a given chromatographic window subjected to MS/MS fragmentation in the linear ion trap. An FTMS target value of $1 \times 10^6$ and an ion trap MSn target value of $1 \times 10^4$ were used with the lock mass (445.120025) enabled. Maximum FTMS scan accumulation time of 100 ms and maximum ion trap MSn scan accumulation time of 50 ms were used. Dynamic exclusion was enabled with a repeat duration of 45 s with an exclusion list of 500 and an exclusion duration of 30 s.

## Mass-spectrometry data analysis

Raw data files were processed using MaxQuant (Version 1.6.10.43) (Tyanova et al, 2016). Data were searched against a combined human and SARS-CoV-2 UniProt sequence database (Dec 2019) using the following search parameters: digestion set to trypsin/P with a maximum of two missed cleavages, oxidation (M), N-terminal protein acetylation as variable modifications, cysteine carbamido-methylation as a fixed modification, match between runs enabled with a match time window of 0.7 and a 20-min alignment time window, label-free quantification was enabled with a minimum ratio count of 2, minimum number of neighbours of 3, and an average number of neighbours of 6. A first search precursor tolerance of 20 ppm and a main search precursor tolerance of 4.5 ppm were used for FTMS scans and a 0.5-D tolerance for ITMS scans. A protein FDR of 0.01 and a peptide FDR of 0.01 were used for identification-level cut-offs.

Protein group output files generated by MaxQuant were loaded into Perseus version 1.6.10.50. The matrix was filtered to remove all proteins that were potential contaminants, only identified by site and reverse sequences. The LFQ intensities were then transformed by log2(x), normalised by subtraction of the median value, and individual intensity columns were grouped by experiment. Proteins were filtered to keep only those with a minimum of three valid values in at least one group. The distribution of intensities was checked to ensure standard distribution for each replicate. Missing values were randomly imputed with a width of 0.3 and downshift of 1.8 from the SD. To identify significant differences between groups, two-sided *t* test were performed with a permutation-based FDR of 0.05.

## Calu-3 cell culture

Calu-3 cells were cultured in DMEM/F12 (1:1) GlutaMAX (Thermo Fisher Scientific) supplemented with 10% FBS, 1% NEAA, 1% sodium pyruvate, 1% penicillin-streptomycin, and maintained at 37°C, 5% $CO_2$, and passaged with TrypLE Express 1X (Thermo Fisher Scientific) when ~80% confluent. All experimental work was performed on cells within the range of 20–30 passages.

## CRISPR-Cas9 editing

An sgRNA targeting exon 1 of EXOSC2 (5′-GAUACAAUCACUACGGACAC-3′) was designed using the CRISPOR tool (http://crispor.tefor.net/) (Concordet & Haeussler, 2018). Design was guided by available protospacer adjacent motifs and predicted on- and off-target efficiencies. A validated, commercially available sgRNA targeting HPRT (IDT) was used as a CRISPR control. sgRNA duplexes were assembled from tracrRNA and crRNA in a thermocycler according to manufacturer's instructions under RNase-free conditions. Cells were cultured to ensure 70–90% confluency on the day of transfection. 24-well plates containing 500 $\mu$l of antibiotic-free (DMEM)/F12 (1:1) GlutaMAX (Thermo Fisher Scientific) were incubated at 37°C. CRISPR/Cas9 ribonucleoproteins were formed by complexing 240 ng sgRNA duplex with 1,250 ng Alt-R V3 Cas9 Protein (IDT) in 10 $\mu$l buffer R (Neon Transfection System 10 $\mu$l Kit; Thermo Fisher Scientific)—a 1:1 M ratio—for 10 min at RT. 100,000 viable cells were aliquoted per transfection and centrifuged at 400$g$ for 5 min at RT. Cells were washed in calcium- and magnesium-free Dulbecco's Phosphate Buffered Saline (Sigma-Aldrich) and centrifuged at 400$g$ for 5 min at RT. Cell pellets were resuspended in 10 $\mu$l buffer R containing Cas9 protein and sgRNA duplexes. 2 $\mu$l of 10.8 $\mu$M electroporation enhancer (IDT) was added, and the solution mixed thoroughly to ensure a suspension of single cells. 10 $\mu$l of this mixture was loaded into a Neon transfection system (Thermo Fisher Scientific) and electroporated according to manufacturer's instructions (1,400 V, 2 pulse, 20 s pulse width). Cells were then transferred to pre-warmed media in 24-well plates.

To assess editing efficiency, genomic DNA was isolated from CRISPR-edited and control cells using a GenElute Mammalian DNA Miniprep Kit (Sigma-Aldrich) according to manufacturer's instructions. A ~400-bp region around the expected Cas9 cut site in exon 1 of EXOSC2 was amplified by PCR using PfuTurbo DNA polymerase (Agilent), according to the manufacturer's instructions using primers: AGGCCGTGAGTTCTCATTGG (fwd) and GGGTTTCAGGGAGCT-GAGAC (rvs) (Sigma-Aldrich). Expected amplification was confirmed using gel electrophoresis, and the products were Sanger-sequenced (Source BioScience). Sequencing trace files were uploaded to TIDE (http://shinyapps.datacurators.nl/tide/) and ICE (https://ice.synthego.com), and an indel efficiency calculated.

## MTT assay

Cell viability was assessed using the MTT Assay Kit (Abcam) according to the manufacturer's instructions. Viable cells but not dead cells can metabolise MTT (3-(4,5-dimethylthiazol-2-yl)-2,5-diphenyltetrazolium bromide) to an insoluble formazan product; when solubilised, the compound can be read at OD590 nm. The measured absorbance is proportional to the number of viable cells. 60,000 Calu-3 cells were seeded in each well of a 96-well plate. After 48 h, the culture medium was discarded, and 50 $\mu$l of fresh serum-free medium and 50 $\mu$l MTT reagent were added to each well,

including to a background (no cells) control. The plates were returned to the 37°C incubator for 3 h before the MTT/media solution was discarded and 150 μl of MTT solvent was added to each well. The plates were incubated for a further 15 min in foil on a hula mixer, and the absorbance recorded at 590 nm. The replicate values were corrected for background and averaged.

### Cloning and viral transduction

The human *EXOSC2* open-reading frame was amplified from HEK293 cDNA using oligonucleotides: gctagcATGGCGATGGAGATGAGGC (fwd) and ctcgagTCCCTCCTGTTCCAAAAGCCT (rvs) and cloned as a NheI/XhoI PCR fragment into the NheI/XhoI restriction sites of a lentiviral self-inactivating transfer vector (SIN) containing a woodchuck hepatitis virus post regulatory element (W) to overexpress EXOSC2 under a PGK promoter (pLV_SIN-W-PGK-EXOSC2). All plasmids were validated by Sanger sequencing, and sequences are available upon request.

HEK293T cells were used for lentiviral production, plated at a density of $3 \times 10^6$ per 10-cm dish. Cells were transfected using a calcium chloride transfection containing 0.5 M calcium chloride (Sigma-Aldrich), 2× HEPES Buffered Saline (Sigma-Aldrich), and four lentiviral component plasmids; pCMV delta 8.2 (13 μg), pRSV-Rev (3 μg), pMD.G (3.75 μg) (Addgene), and pLV_SIN-W-PGK-EXOSC2 (13 μg) (Déglon et al, 2000). Transfection mix added dropwise to each plate and left overnight, with a full media change carried out the following morning. Cells were incubated for a further 48 h before all media was collected and filtered using a 0.45-μm filter (Sigma-Aldrich). Equally loaded tubes (Beckman Coulter) were then spun at 19,000 rpm/90 min/4°C using an ultracentrifuge and a SW28 hanging rota (Beckman Coulter). All supernatant was removed, and each viral pellet was resuspended with 300 μl of 1% bovine serum albumin (Tocris Bioscience) in Phosphate Buffer Solution (Sigma-Aldrich). Each tube was incubated on ice for 1 h and then combined into one homogeneous solution before being aliquoted and stored at –80°C.

Viral titres were measured through qPCR against a virus of known biological titre (FACS titration). Genomic DNA was isolated from cells (described previously) which had been transduced with a serial dilution of virus. Viral genomic integration was measured using WPRE primers: CCCGTACGGCTTTCGTTTTC (fwd) and CAAACA-CAGAGCACACCACG (rvs).

### SARS-CoV-2 production

The SARS-CoV-2 strain Victoria was produced by infecting Vero E6 cells at an MOI of 0.01 in DMEM supplemented with 2% FCS for 72 h, until cytopathic effects were visible. The supernatant containing viral particles was harvested, aliquoted, and stored at –80°C. The titre of the SARS-CoV-2 stock was determined by plaque assay using Vero E6 cells.

### SARS-CoV-2 challenge

1 million Calu-3 cells were seeded in each well of a 12-well plate. Cells were left to adhere for 24 h. The cells were infected with the SARS-CoV-2 strain Victoria at MOI 1 in 2% FCS-medium, allowing a total volume of 1 ml per well. The cells were returned to the incubator and harvested at 17 h post-infection when 20% of cells showed

cytopathogenic effects. To neutralise SARS-CoV-2, 1 ml of SARS-CoV-2 was incubated for 1 h before infection with 15 μl of the antibody clone EY11A with regular mixing. EY11A inhibits SARS-CoV-2 activity by binding strongly to its spike protein. At the end point, the supernatant was collected and preserved at –80°C for future determination of virus titre, and after two washes with PBS, RNA was extracted from cells using the RNeasy Mini Kit (QIAGEN) following manufacturer's instructions.

For the RNA sequencing experiment, 2 million cells were seeded in six-well plates, and after 24 h, cells were infected with THE SARS-CoV-2 Victoria strain at MOI 1. At 17 h post infection, cells were washed twice with PBS and harvested for RNA extraction using the QIAGEN RNeasy Mini Kit. The quality of the RNA was assessed by NanoDrop ratios and TapeStation 4200 (Agilent). RNAs with a 260/230 ratio > 2 and RIN > 9.8 were submitted for library preparation and sequencing.

### TCID50 titration of virus after challenge

96-well plates were seeded with 15,000 Vero E6 cells per well 24 h before adding virus in 120 μl of 2% FCS medium per well. 40 μl/well of preserved supernatant was added to the monolayer of Vero E6 cells in column 1 of the plate (rows A to H); this provides eight replicates to determine TCID50/ml. The virus was mixed by carefully pipetting six times before changing tips at each column and transferring 40 μl to the next column, performing a serial dilution (1:4) across the plate. The plates were incubated at 37°C for 72 h and stained with crystal violet. TCID50/ml was determined visually by recording cell death at each dilution and deriving the titre using the freely available online software (https://www.klinikum.uni-heidelberg.de/fileadmin/inst_hygiene/molekulare_virologie/Downloads/TCID50_calculator_v2_17-01-20_MB.xlsx).

### Quantification of the SARS-CoV-2 RNA copy number

The SARS-CoV-2 RNA copy number was quantified in RNA samples extracted from infected cells using the QuantiTect probe RT-PCR Kit (QIAGEN). We prepared a standard curve using the ORF9 nucleoprotein coding sequence cloned in pcDNA3. 10 μg of plasmid was linearised with Apa1 at 37°C for 2 h. DNA was purified with the PCR cleanup kit from QIAGEN, and 1 μg of DNA was used in a T7 transcription reaction using Promega T7 RiboMAX (P1320) at 37°C for 30 min. The transcribed RNA was cleaned using Zymo RNA clean and concentrator 25 (R1017; Zymo Research). The copy number was derived using an online calculator (https://nebiocalculator.neb.com/ssRNA mass to moles converter).

The RNA standard curve was prepared from diluting RNA template from $10 \times 10^8$ to $10 \times 10^1$ in RNase-free water. RNA samples were diluted 1:5,000, and 1 μl of standard or sample was dispensed into each well of a 384-well plate to perform the RT-PCR step following manufacturer's instructions. Quantitation of the copy number was performed using N1 and N2 standard probes for SARS-CoV-2 (IDTDNA).

### EXOSC2 reconstitution

5 μg of pLenti-EXOSC2 with 2.5 μg pR8.91 and 2.5 μg of VSV-G plasmids were transfected in a 15-cm dish of Hek293T cells at

80% confluency. The transfection mix was prepared using Fugene 6 (Promega) at 3 µl/µg DNA in 500 µl (total volume) of Opti-MEM (Thermo Fisher Scientific). The transfection mix was incubated following manufacturer's recommendations and added dropwise to the dish. After 24 h, the medium was changed to 15 ml 2% FCS medium. After a further 24-h incubation, the supernatant was collected, spun at 200$g$ to remove cellular debris, and passed through 0.45-µM syringe filter (Sartorius). The supernatants were preserved at –80°C.

To transduce cells with EXOSC2-lentivirus, a T25 of EXOSC2 CRISPR-edited cells was incubated for 48 h with 2 ml of culture media supplement with 1 ml of viral supernatant and 10 µg of polybrene. The cells were then returned to normal propagation media.

## Immunoblotting for EXOSC2

500,000 Calu-3 cells were seeded in 12-well plates and left to adhere for 24 h. Then the media was discarded, cells were subsequently washed with PBS, and 250 µl of RIPA buffer (10 mM Tris–HCl pH 8, 140 mM NaCl, 1% Triton-X100, 0.1% SDS, 0.1% sodium deoxycholate, 1 mM EDTA, 1 mM EGTA) containing protease inhibitor (04693132001; Roche) was added to the cells. Cells were scraped from wells, dispensed into an Eppendorf tube, and agitated for 20 min at 4°C, then spun at 30,000$g$ for 20 min at 4°C. The supernatant was collected, and 100 µg of total protein (determined by qubit) was supplemented with Laemmli buffer, boiled for 5 min, and resolved on 4–20% TGX mini protean gels (Bio-Rad). The proteins were transferred to PVDF membranes using Bio-Rad Trans-Blot semi-dry transfer. The membrane was blocked for 1 h with constant rotation in 2.5% milk-PBS 0.1% Tween 20 and incubated overnight at 4°C with the EXOSC2 antibody (66099-1-Ig; ProteinTech Group) or GAPDH antibody (14C10; Cell Signalling Technology) diluted in 2.5% milk-PBS-0.1% Tween 20. After three washes of 15 min each in PBS-0.1% Tween 20, the membranes were incubated with secondary antibodies for 1 h, and after three further washes with PBST, the membrane was transferred to a fresh falcon tube and incubated with 1 ml of Western Lightning Plus ECL chemiluminescent reagent (PerkinElmer). Proteins bands were visualised with an iBright instrument (Thermo Fisher Scientific). The intensity of EXOSC2 band against the GAPDH loading control was assessed using ImageJ.

## Immunoblotting for the total set of RNA exosome proteins

Calu-3 cells were harvested by centrifugation at 400$g$ for 4 min at RT. Cell pellets were washed with ice-cold PBS and lysed in RIPA buffer (10 mM Tris–HCl, pH 8, 140 mM NaCl, 1% Triton-X100, 0.1% SDS, 0.1% sodium deoxycholate, 1 mM EDTA, 1 mM EGTA) containing protease inhibitor cocktail (Roche) for 15 min on ice. Lysates were centrifuged at 17,000$g$ for 5 min at 4°C, and the supernatant was transferred to fresh Eppendorf tubes. 30 µg total protein lysate was mixed with Laemmli buffer, boiled at 95°C, and fractionated on 12% SDS polyacrylamide gels. Protein was electrophoretically transferred to nitrocellulose membranes using a Bio-Rad semi-dry transfer apparatus. Membranes were blocked in 5% milk + TBST (20 mM Tris, 137 mM NaCl, 0.2% [vol/vol] Tween 20, pH 7.6) for 1 h at RT, then incubated overnight at 4°C in the primary antibody (EXOSC1,

12585-1-AP; Proteintech; EXOSC2, 14805-1-AP; Proteintech; EXOSC3, 15062-1-AP; Proteintech; EXOSC4, 15937-1-AP; Proteintech; EXOSC5, 15627-1-AP; Proteintech; EXOSC7, 25292-1-AP; Proteintech; EXOSC8, 11979-1-AP; Proteintech; EXOSC9, 24470-1-AP; Proteintech; EXOSC10, 11178-1-AP; Proteintech; DIS3, 14689-1-AP; Proteintech; DIS3L, 25746-1-AP; Proteintech; DIS3L2, 25792-1-AP; Proteintech; β-actin, ab8227; Abcam) diluted in 5% milk + TBST. Membranes were washed 3× in TBST then incubated in an HRP-conjugated secondary antibody (diluted in 5% milk + TBST) for 1 h at RT, followed by a further three washes in TBST. Membranes were incubated in ECL Western blotting substrates (Bio-Rad) for 5 min at RT, and protein bands were visualised using an LI-COR Odyssey XF Imaging System. Quantification of band intensity was performed using FIJI (NIH) and normalised to β-actin.

## qRT–PCR for EXOSC2

Calu-3 cells were lysed on ice using Tri Reagent (Sigma-Aldrich) for 5 min in RNase-free conditions. Total RNA was extracted using a Direct-zol RNA Miniprep Kit (Zymo) according to the manufacturer's instructions. RNA concentration was determined using a NanoDrop spectrophotometer (Thermo Fisher Scientific). 2 µg total RNA was converted to cDNA by adding 1 µl 10 mM dNTPs, 1 µl 40 µM random hexamer primer, and DNAse/RNAse–free water to a total reaction volume of 14 µl. The mixture was heated to 70°C followed by a 5-min incubation on ice. 4 µl 5× first strand buffer, 2 µl 0.1 M DTT, and 1 µl M-MLV reverse transcriptase (Thermo Fisher Scientific) were then added, and cDNA conversion was performed in a thermocycler (37°C for 50 min, 70°C for 10 min). cDNA was amplified using RT-PCR with Brilliant III SYBR Green (Agilent) as per the manufacturer's instructions using primers: AACCTGGAGCCTGTCTCTCTT (fwd) and TGATCTGATGTGGAAGGGATGC (rvs). CT analysis was performed using CFX Maestro software (Bio-Rad).

## RNA sequencing

RNA was extracted from Calu-3 cells in the presence and absence of *EXOSC2* gene editing at 17 h post–SARS-CoV-2 infection at MOI 1. Extracted RNA was of high quality (RIN~10); the input mass was 100 ng. Libraries were prepared according to manufacturer's instructions; the NEBNext rRNA Depletion Kit v2 was applied to remove human ribosomal RNA before strand-specific libraries were constructed using the NEBNext Ultra II Directional RNA Library Prep Kit for Illumina (E7760). The libraries were indexed with custom adaptors and barcode tags (including dual indexing [Lamble et al, 2013]) and sequenced on an Illumina NovaSeq6000 v1.5 in 150-bp paired-end mode. A minimum of 80 million reads were obtained per sample.

## Human transcriptome analysis

Raw Fastq files were trimmed for the presence of Illumina adaptor sequences using Cutadapt v1.2.1 (Martin, 2011). Reads were aligned to hg19 transcripts ($n$ = 180,253) using Kallisto v0.46.0 (Bray et al, 2016) to produce gene-level TPM estimates by aggregating transcripts per gene. Differential expression analysis was performed using edgeR (Robinson et al, 2010). Read counts were first TMM-normalised (Robinson & Oshlack, 2010) to account for differences in

library size. Differentially expressed genes were identified by a genewise exact test (Robinson & Smyth, 2008); only genes which were significant after Benjamini–Hochberg multiple testing correction were reported as differentially expressed.

## Viral transcriptome analysis

The raw fastQ files were filtered for viral reads using ReadItAndKeep (Hunt et al, 2022), and the files of viral reads produced were then used for downstream analysis of Subgenomic RNA. Briefly, ReadItAndKeep removes host reads from viral sequencing data by aligning all reads against a target genome using minimap2 and retaining only reads that match at least 50 bp or 50% of the length of the read. This approach has been shown to have 100% sensitivity and 99.894% specificity in distinguishing human from viral reads with Illumina sequencing. ReadItAndKeep version 0.1.0 was downloaded from Bioconda and run using the Wuhu-Hu-1 genome (NCBI Reference Sequence: NC_045512.2, download from www.ncbi.nlm.nih.gov/nuccore/NC_045512) as the target sequence the paired Illumina reads in fastq format. The percentage of viral reads was calculated as the number of reads retained by ReadItAndKeep divided by the total number of reads.

To calculate the proportion of genomic:subgenomic reads, we measured read-depth over SARS-CoV-2 Orf1 which is present in all genomic RNAs and none of the canonical subgenomic RNAs to the read-depth over Orf10 which is present in all canonical subgenomic RNAs (Long, 2021). A limitation of our method is that we do not consider low-frequency non-canonical subgenomic RNAs.

# Data Availability

The mass spectrometry data have been deposited to the ProteomeXchange Consortium via the PRIDE partner repository with the dataset identifier PXD031611.

# Supplementary Information

# Acknowledgements

This work was supported by the National Institutes of Health (CEGS 5P50HG00773504, 1P50HL083800, 1R01HL101388, 1R01-HL122939, S10OD025212, P30DK116074, and UM1HG009442 to MP Snyder), the Wellcome Trust (216596/Z/19/Z to J Cooper-Knock), the NIHR (NF-SI-0617-10077 to PJ Shaw), the BBSRC (BB/S009566/1 to A Peden and MO Collins), and the UK Medical Research Council (MRC core funding of the MRC Human Immunology Unit to J Rehwinkel). C Harvey is supported by a studentship from the MND Association (899-792). We also acknowledge support from a Kingsland fellowship (T Moll) and the NIHR Sheffield Biomedical Research Centre for Translational Neuroscience (IS-BRC-1215-20017). We thank the Oxford Genomics Centre at the Wellcome Centre for Human Genetics (funded by Wellcome Trust grant reference 203141/Z/16/Z) for the generation and initial processing of sequencing data. Fig 1A was created with BioRender.com. The neutralising antibody was a kind gift from Alain Townsend.

## Author Contributions

T Moll: conceptualization, data curation, formal analysis, investigation, methodology, and writing—original draft.
V Odon: data curation, formal analysis, validation, investigation, visualization, methodology, and writing—original draft.
C Harvey: data curation, software, formal analysis, validation, investigation, and visualization.
MO Collins: data curation, software, formal analysis, supervision, funding acquisition, investigation, methodology, project administration, and writing—original draft, review, and editing.
A Peden: data curation, formal analysis, investigation, methodology, and project administration.
J Franklin: formal analysis, investigation, and methodology.
E Graves: data curation, investigation, and methodology.
JNG Marshall: formal analysis, investigation, and methodology.
C dos Santos Souza: formal analysis, investigation, and methodology.
S Zhang: conceptualization, formal analysis, investigation, and methodology.
L Castelli: data curation and investigation.
G Hautbergue: data curation and investigation.
M Azzouz: investigation, methodology, and project administration.
D Gordon: investigation and methodology.
N Krogan: investigation and methodology.
L Ferraiuolo: investigation and methodology.
MP Snyder: investigation and methodology.
PJ Shaw: supervision, investigation, methodology, and project administration.
J Rehwinkel: conceptualization, resources, data curation, formal analysis, supervision, funding acquisition, validation, investigation, visualisation, methodology, project administration, and writing—original draft, review, and editing.
J Cooper-Knock: conceptualization, data curation, software, formal analysis, supervision, funding acquisition, investigation, visualisation, methodology, project administration, and writing—original draft, review, and editing.

## Conflict of Interest Statement

MP Snyder is a co-founder and member of the scientific advisory board of Personalis, Qbio, January, SensOmics, Protos, Mirvie, NiMo, Onza and Oralome. He is also on the scientific advisory board of Danaher, Genapsys and Jupiter. The Krogan Laboratory has received research support from Vir Biotechnology, F Hoffmann-La Roche, and Rezo Therapeutics. Nevan Krogan has financially compensated consulting agreements with the Icahn School of Medicine at Mount Sinai, New York, Maze Therapeutics, Interline Therapeutics, Rezo Therapeutics, GEn1E Lifesciences, Inc. and Twist Bioscience Corp. He is on the Board of Directors of Rezo Therapeutics and is a shareholder in Tenaya Therapeutics, Maze Therapeutics, Rezo Therapeutics, and Interline Therapeutics.

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
