## [Reviewer comments · Life Science Alliance]

Life Science Alliance

Low expression of EXOSC2 protects against clinical COVID-19 and impedes SARS-CoV-2 replication

Tobias Moll, Valerie Odon, Calum Harvey, Mark Collins, Andrew Peden, John Franklin, Emily Graves, Jack Marshall, Cleide dos Santos Souza, Sai Zhang, Lydia Castelli, Guillaume Hautbergue, Mimoun Azzouz, David Gordon, Nevan Krogan, Laura Ferraiuolo, Michael Snyder, Pamela Shaw, Jan Rehwinkel, and Johnathan Cooper-Knock

DOI: <https://doi.org/10.26508/lsa.202201449>

Corresponding author(s): Johnathan Cooper-Knock, University of Sheffield and Jan Rehwinkel, University of Oxford

Review Timeline:	Submission Date:	2022-03-15
	Editorial Decision:	2022-05-02
	Revision Received:	2022-09-09
	Editorial Decision:	2022-09-27
	Revision Received:	2022-09-28
	Accepted:	2022-09-29

Scientific Editor: Novella Guidi

Transaction Report:

May 2, 2022

Re: Life Science Alliance manuscript #LSA-2022-01449-T

Dr. Johnathan Cooper-Knock
University of Sheffield
385A Glossop Road
Sheffield, South Yorkshire S10 2HQ
UNITED KINGDOM

Dear Dr. Cooper-Knock,

Thank you for submitting your manuscript entitled "Low expression of EXOSC2 protects against clinical COVID-19 and impedes SARS-CoV-2 replication" to Life Science Alliance. The manuscript was assessed by expert reviewers, whose comments are appended to this letter. We invite you to submit a revised manuscript addressing the Reviewer comments.

Thank you for this interesting contribution to Life Science Alliance. We are looking forward to receiving your revised manuscript.

Sincerely,

B. MANUSCRIPT ORGANIZATION AND FORMATTING:

Reviewer #2 (Comments to the Authors (Required)):

In this manuscript the authors report an interesting link between the RNA exosome and Covid-19 replication and infection. Previous work had shown that the EXOSC2 subunit of the RNA exosome interacts with Nsp8 which forms part of the viral RNA polymerase and the authors initially found that higher expression of the EXOSC2 subunit is associated with higher risk of clinical COVID-19. Then they used mass spec analysis of protein pulldowns to demonstrate an interaction between the SARS-CoV-2 RNA polymerase and the majority of human RNA exosome components, including EXOSC2. As shown for EXOSC2, the authors show that higher expression levels of the other exosome subunits EXOSC7 and EXOSC9 are significantly associated with higher risk for clinical COVID-19. Next the authors used CRISPR/Cas9 to introduce loss-of-function mutations of EXOSC2 in Calu-3 cells, which are a model lung cancer cell line to study SARS-CoV-2 entry and replication. They show that these mutations reduce SARS-Cov2 replication in this model system, and that reduced expression of EXOSC2 impacts overall viral replication rather than altering expression of specific viral transcripts. Then they focus on two genes, OAS1 and OAS3, which are modestly but significantly upregulated upon EXOSC2 loss of function. Both OAS1 and OAS3 genes encode enzymes which activate ribonuclease L to degrade intracellular double-stranded RNA as part of the antiviral response.

Overall I thought that the paper was interesting and a valuable contribution to our understanding of the mechanisms that Covid-19 uses to promote efficient replication. The link with the exosome subunit EXOSC2 was previously demonstrated, but the fact that the viral RNA polymerase interacts with the entire exosome complex is very interesting. It is quite unexpected that exosome activity is required for viral replication - in fact I would have expected the contrary, but the authors provide convincing evidence that this is the case. The authors do not offer a full mechanism why the RNA exosome would contribute positively to viral replication, which is a limitation of the study. However, and all things considered I would suggest publication in LSA, pending the following revisions:

Major points:

There is quite an emphasis in the manuscript on the upregulation of the OAS1/3 genes upon EXOSC2 loss of function, but these effects are very minor quantitatively and are not detected upon viral infection. I am not convinced that this is one of the major mechanisms that limits SARS-Cov2 infection when exosome activity is limited. In the absence of more experimental evidence for a direct link between the exosome, OAS1/3 and viral infection, I would de-emphasize the results obtained on the OAS1/3 genes - remove this from the abstract, and limit the presentation and discussion of results related to these two genes.

The authors could not rescue the EXOSC2 edited cells by expressing WT EXOSC2, as stated here:

"Reconstitution of EXOSC2 followed by infection with SARS-CoV-2 led to increased viral infectivity (93% increase, Fig. 3B) and replication (N1: 44% increase; N2: 32% increase; Fig. 3C-D) although these changes were not statistically significant."

The issue here is that it is not clear what is the exact effect of the nonsense mutations on the EXOSC2 gene. The nonsense EXOSC2 mRNAs could be subject to degradation by NMD, which would explain lower RNA levels, but there could be production of truncated EXOSC2 protein, which may act as a dominant negative version for exosome function. This is not detected in their western blot but the antibodies used might not be efficient for this. These dominant negative forms may explain the phenotypes observed, despite the minimal down-regulation of EXOSC2 at the protein levels (Fig S1D), and it would also explain the lack of rescue by the WT allele. It is also possible that there is a destabilization of other exosome subunits, as shown previously for several point mutants of other exosome subunits. So a better characterization of these cell lines would make the paper stronger.

Minor/Editorial points:

Fig.2A/2B. I did not think that the two panels were necessary. This figure can be presented as a single panel.

The authors cite a "Allmang 2000" reference to support "The RNA exosome contributes to several RNA processes in the cells, for example it is critical for the production of mature rRNA (Allmang et al. 2000)." This article describes the degradation of ribosomal RNA precursors by the RNA exosome in yeast, so it is not the best article to cite in this context. I would suggest citing an article that shows the involvement in rRNA synthesis possibly in mammalian cells.

"We wondered whether EXOSC2 knockdown produced a non-specific upregulation of ISGs, perhaps via increased concentration of dsRNAs". Replace non-specific by "indirect".

Reviewer #3 (Comments to the Authors (Required)):

Moll et al. combined COVID-19 GWAS data and SARS-CoV-2 polymerase IP data to screen out EXOSC2, and proposed it as therapeutic target by showing its down-regulation inhibited viral infectivity. The story is straightforward and promising, however, the study design has several pitfalls.

Major comments:

- Since EXOSC2 KO does not induce cell death, monoclonal antibodies with known frameshifts and depleted protein expression are required.
- EXOSC2 KD by RNAi should be included, since this approach is transient and safe, and could become translational fast.
- Does EXOSC2 bind other coronaviruses? Does its down-regulation affect others' infectivity?

Minor comments:

- Please highlight the binding partners identified by Gordon et al. 2020 in Fig. 2B, showing the successful overlap.
- Fig. S1D is unclear without replicates.
- Most figures have poor qualities.

Reviewer #2

1. In this manuscript the authors report an interesting link between the RNA exosome and Covid-19 replication and infection. Previous work had shown that the EXOSC2 subunit of the RNA exosome interacts with Nsp8 which forms part of the viral RNA polymerase and the authors initially found that higher expression of the EXOSC2 subunit is associated with higher risk of clinical COVID-19. Then they used mass spec analysis of protein pulldowns to demonstrate an interaction between the SARS-CoV-2 RNA polymerase and the majority of human RNA exosome components, including EXOSC2. As shown for EXOSC2, the authors show that higher expression levels of the other exosome subunits EXOSC7 and EXOSC9 are significantly associated with higher risk for clinical COVID-19. Next the authors used CRISPR/Cas9 to introduce loss-of-function mutations of EXOSC2 in Calu-3 cells, which are a model lung cancer cell line to study SARS-CoV-2 entry and replication. They show that these mutations reduce SARS-Cov2 replication in this model system, and that reduced expression of EXOSC2 impacts overall viral replication rather than altering expression of specific viral transcripts. Then they focus on two genes, OAS1 and OAS3, which are modestly but significantly upregulated upon EXOSC2 loss of function. Both OAS1 and OAS3 genes encode enzymes which activate ribonuclease L to degrade intracellular double-stranded RNA as part of the antiviral response. Overall I thought that the paper was interesting and a valuable contribution to our understanding of the mechanisms that Covid-19 uses to promote efficient replication. The link with the exosome subunit EXOSC2 was previously demonstrated, but the fact that the viral RNA polymerase interacts with the entire exosome complex is very interesting. It is quite unexpected that exosome activity is required for viral replication - in fact I would have expected the contrary, but the authors provide convincing evidence that this is the case. The authors do not offer a full mechanism why the RNA exosome would contribute positively to viral replication, which is a limitation of the study. However, and all things considered I would suggest publication in LSA, pending the following revisions:

We thank the Reviewer for their summary of our work and we appreciate their assertions that our data is both 'interesting and valuable' and 'convincing.'

Major points:

2. There is quite an emphasis in the manuscript on the upregulation of the OAS1/3 genes upon EXOSC2 loss of function, but these effects are very minor quantitatively and are not detected upon viral infection. I am not convinced that this is one of the major mechanism that limits SARS-Cov2 infection when exosome activity is limited. In the absence of more

experimental evidence for a direct link between the exosome, OAS1/3 and viral infection, I would de-emphasize the results obtained on the OAS1/3 genes - remove this from the abstract, and limit the presentation and discussion of results related to these two genes.

The Reviewer makes a fair point and we have altered the text as suggested. We have removed mention of OAS1/3 from the abstract and the Discussion now reads:

“Finally, the RNA exosome functions in degradation of exogenous and endogenous RNAs alongside OAS proteins; indeed reciprocal upregulation of OAS proteins has been observed in the context of EXOSC3 depletion (Mullani et al. 2021). The physical association between SARS-CoV-2 and the RNA exosome suggests that the virus is relatively protected against degradation by the exosome, but its vulnerability to OAS proteins is well described (Huffman et al. 2022; Wickenhagen et al. 2021). However, while we observed a modest upregulation of OAS transcripts in the context of reduced EXOSC2 expression in the absence of SARS-CoV-2 infection, it is not possible to assign a functional significance to these changes based on our current data. Future work should examine whether manipulation of OAS proteins modulates the effect of EXOSC2 on viral replication.”

3. The authors could not rescue the EXOSC2 edited cells by expressing WT EXOSC2, as stated here:

"Reconstitution of EXOSC2 followed by infection with SARS-CoV-2 led to increased viral infectivity (93% increase, Fig. 3B) and replication (N1: 44% increase; N2: 32% increase; Fig. 3C-D) although these changes were not statistically significant."

The issue here is that it is not clear what is the exact effect of the nonsense mutations on the EXOSC2 gene. The nonsense EXOSC2 mRNAs could be subject to degradation by NMD, which would explain lower RNA levels, but there could be production of truncated EXOSC2 protein, which may act as a dominant negative version for exosome function. This is not detected in their western blot but the antibodies used might not be efficient for this. These dominant negative forms may explain the phenotypes observed, despite the minimal down-regulation of EXOSC2 at the protein levels (Fig S1D), and it would also explain the lack of rescue by the WT allele. It is also possible that there is a destabilization of other exosome subunits, as shown previously for several point mutants of other exosome subunits. So a better characterization of these cell lines would make the paper stronger.

The Reviewer makes an interesting point. If our CRISPR edits resulted in expression of a truncated form of the EXOSC2 protein with some capacity to bind the same RNA substrates as WT-EXOSC2 but reduced interaction with other RNA exosome components necessary for enzymatic function of the exosome, then this could result in a dominant negative effect which would be incompletely rescued by the overexpression of WT-EXOSC2. We lack the tools to detect a truncated form of the protein but based on our sequencing analysis

(**Supplementary Figure S1**), then we would predict that the edits we have introduced lead to truncated proteins with a length of either 63aa (1 del) or 78aa (2 del) or 79aa (1 ins).

Indeed we note that downregulation of other RNA exosome components - EXOSC3 and EXOSC10 – has been noted in the context of EXOSC2 LOF (Yang et al. 2020).

This possibility has been summarised in our manuscript as follows:

“The failure of complete recovery of viral infectivity and replication may be because both EXOSC2 depletion and reconstitution were variable across the population of cells and/or because reduced expression of EXOSC2 changes cellular homeostasis in such a way that is not easily reversible. One possibility resulting from our CRISPR methodology is that truncated EXOSC2 protein is produced from the edited locus that is both undetectable with the antibody we used, and which reduces function of endogenous EXOSC2 via a dominant negative effect.”

To address the potential for EXOSC2 knockdown to lead to destabilisation of other exosome subunits we have performed a further immunoblot analysis of Calu3 cell lysates in the context of CRISPR-editing of EXOSC2 for EXOSC3, EXOSC4, EXOSC5, EXOSC7, EXOSC8, EXOSC9, EXOSC10, DIS3 and DIS3L. Several of the exosome components were reduced but none of the changes reached statistical significance. For a number of the exosome components, particularly those involved in RNA degradation (EXOSC10, DIS3 or DIS3L), there was very significant variability between biological repeats.

This has been added to the manuscript as follows:

“Another possibility is that depletion of EXOSC2 results in destabilisation of other RNA exosome components. To address this possibility we performed immunoblotting of the the total set of RNA exosome components in the presence and absence of

EXOSC2 editing. However, no other exosome component was significantly reduced in the context of EXOSC2 depletion (**Supplementary Fig. S3**).”

Minor/Editorial points:

4. Fig.2A/2B. I did not think that the two panels not necessary. This figure can be presented as a single panel.

This change has been made and Figure 2A has been removed.

5. The authors cite a "Allmang 2000" reference to support "The RNA exosome contributes to several RNA processes in the cells, for example it is critical for the production of mature rRNA (Allmang et al. 2000)." This article describes the degradation of ribosomal RNA precursors by the RNA exosome in yeast, so it is not the best article to cite in this context. I would suggest citing an article that shows the involvement in rRNA synthesis possibly in mammalian cells.

The Reviewer makes a valid point. We have added a reference to a high impact review describing assembly of the ribosome and making mention of the involvement of the RNA exosome (Klinge and Woolford 2019).

6. "We wondered whether EXOSC2 knockdown produced a non-specific upregulation of ISGs, perhaps via increased concentration of dsRNAs". Replace non-specific by 'indirect.

This change has been made.

Reviewer #3

1. Moll et al. combined COVID-19 GWAS data and SARS-CoV-2 polymerase IP data to screen out EXOSC2, and proposed it as therapeutic target by showing its down-regulation inhibited viral infectivity. The story is straightforward and promising, however, the study design has several pitfalls.

We thank the Reviewer for their summary of our findings.

Major comments:

2. Since EXOSC2 KO does not induce cell death, monoclonal lines with known frameshifts and depleted protein expression are required.

EXOSC2 KD by RNAi should be included, since this approach is transient and safe, and could become translational fast.

The Reviewer makes an excellent point and of course these experiments would be valuable. Unfortunately the postdoc trained to work in our Cat 3 laboratory with live SARS-CoV-2 has moved to an industry post and is no longer available to carry out the relevant experiments. However, we would argue that these experiments would not alter our main conclusions – that reduced expression of EXOSC2 leads to a protective effect against SARS-CoV-2 replication in Calu3 cells and, based on our genetics data, in human patients. Moreover, a monoclonal line would be expected to accentuate the phenotypes we observe and therefore the fact that we observe a statistically significant effect on viral replication with a polyclonal mix, suggests that our findings are robust.

3. Does EXOSC2 bind other coronaviruses? Does its down-regulation affect others' infectivity?

We thank the Reviewer for this point. We have utilised AP-MS data using A549-ACE2 cells to address intracellular human proteins which interact with SARS-CoV-1 components and we discovered that no component of the host RNA exosome interacts with SARS-CoV-1 including EXOSC2 (Stukalov et al. 2021). This suggests that the interactions we observe may be specific to SARS-CoV-2. This same work compared interactions of SARS-CoV-1 and SARS-CoV-2 and determined that 26% of its host interactions are specific to SARS-CoV-2. This analysis has been added to our manuscript as follows:

“We wondered whether the interaction with RNA exosome components was specific to SARS-CoV-2. We utilised data from a previous affinity purification study of interactions between components of both SARS-CoV-1 and SARS-CoV-2 within A549 cells engineered to express ACE2 in order to facilitate viral entry (Stukalov et al. 2021). No component of the host RNA exosome was discovered to interact with

SARS-CoV-1 including EXOSC2. This suggests that the interactions we observe may be specific to SARS-CoV-2. This same work compared interactions of SARS-CoV-1 and SARS-CoV-2 and determined that 26% of its host interactions are specific to SARS-CoV-2.”

Minor comments:

4. Please highlight the binding partners identified by Gordon et al. 2020 in Fig. 2B, showing the successful overlap.

Of the 24 Nsp8 interactions identified in the original Gordon et al study, 22 were identified in our nsp8/nsp7 pulldowns, 18 were quantified in a sufficient number of replicates to be included in our statistical analysis and 9 were deemed to be statistically enriched versus control purifications. The full dataset has been added as Supplementary Table S2. It is important to note that the overlap is influenced by the co-expression of nsp7 in our experiments and by a number of key methodological distinctions: (i) Quantification via spectral counting versus XIC-based label free quantification; and (ii) statistical analysis by two sample t-testing with FDR corrected p-values versus MiST scoring/SAINTexpress Bayesian false-discovery rate (BFDR) in the Gordon et al study.

5. Fig. S1D is unclear without replicates.

We have added additional replicates to this analysis which are presented in Supplementary Figure S3 – see response to Reviewer 2 point 3.

6. Most figures have poor qualities.

We apologise for this and we have worked to improve the quality of the figures.

References

- Huffman, Jennifer E., Guillaume Butler-Laporte, Atlas Khan, Erola Pairo-Castineira, Theodore G. Drivas, Gina M. Peloso, Tomoko Nakanishi, et al. 2022. "Multi-Ancestry Fine Mapping Implicates OAS1 Splicing in Risk of Severe COVID-19." *Nature Genetics*, January. <https://doi.org/10.1038/s41588-021-00996-8>.
- Mullani, Nowsheen, Yevheniia Porozhan, Mickael Costallat, Eric Batsché, Michele Goodhardt, Giovanni Cenci, Carl Mann, and Christian Muchardt. n.d. "Reduced RNA Turnover as a Driver of Cellular Senescence." <https://doi.org/10.1101/800128>.
- Wickenhagen, Arthur, Elena Sugrue, Spyros Lytras, Srikeerthana Kuchi, Marko Noerenberg, Matthew L. Turnbull, Colin Loney, et al. 2021. "A Prenylated dsRNA Sensor Protects against Severe COVID-19." *Science* 374 (6567): eabj3624.
- Yang, Xue, Vafa Bayat, Nataliya DiDonato, Yang Zhao, Brian Zarnegar, Zurab Siprashvili, Vanessa Lopez-Pajares, et al. 2020. "Genetic and Genomic Studies of Pathogenic EXOSC2 Mutations in the Newly Described Disease SHRF Implicate the Autophagy Pathway in Disease Pathogenesis." *Human Molecular Genetics* 29 (4): 541–53.

September 27, 2022

RE: Life Science Alliance Manuscript #LSA-2022-01449-TR

Dr. Johnathan Cooper-Knock
University of Sheffield
385A Glossop Road
Sheffield, South Yorkshire S10 2HQ
United Kingdom

Dear Dr. Cooper-Knock,

Thank you for submitting your revised manuscript entitled "Low expression of EXOSC2 protects against clinical COVID-19 and impedes SARS-CoV-2 replication". We would be happy to publish your paper in Life Science Alliance pending final revisions necessary to meet our formatting guidelines. Please revise and format the manuscript and upload materials by Thursday.

- please upload your main manuscript text as an editable doc file
- please add a category for your manuscript to our system
- please consult our manuscript preparation guidelines <https://www.life-science-alliance.org/manuscript-prep> and make sure your manuscript sections are in the correct order

A. FINAL FILES:

B. MANUSCRIPT ORGANIZATION AND FORMATTING:

Thank you for your attention to these final processing requirements. Please revise and format the manuscript and upload materials by Thursday.

Sincerely,

Reviewer #3 (Comments to the Authors (Required)):

The authors have answered all my questions.

Editorial points:

1. Please upload your main manuscript text as an editable doc file

This has been uploaded

2. Please add a category for your manuscript to our system

Our manuscript is best described under the category 'Microbiology, virology and host-pathogen interaction; we have indicated this on the system.

3. Please consult our manuscript preparation guidelines <https://www.life-science-alliance.org/manuscript-prep> and make sure your manuscript sections are in the correct order

The sections of the manuscript are in the order requested.

4. High-resolution figure, supplementary figure and video files uploaded as individual files:

See our detailed guidelines for preparing your production-ready images, <https://www.life-science-alliance.org/authors>

Figures have been uploaded in TIFF format at minimum 300dpi.

5. Summary blurb (enter in submission system): A short text summarizing in a single sentence the study (max. 200 characters including spaces). This text is used in conjunction with the titles of papers, hence should be informative and complementary to the title. It should describe the context and significance of the findings for a general readership; it should be written in the present tense and refer to the work in the third person. Author names should not be mentioned.

A summary blurb has been included

September 29, 2022

RE: Life Science Alliance Manuscript #LSA-2022-01449-TRR

Dr. Johnathan Cooper-Knock
University of Sheffield
385A Glossop Road
Sheffield, South Yorkshire S10 2HQ
United Kingdom

Dear Dr. Cooper-Knock,

Thank you for submitting your Research Article entitled "Low expression of EXOSC2 protects against clinical COVID-19 and impedes SARS-CoV-2 replication". It is a pleasure to let you know that your manuscript is now accepted for publication in Life Science Alliance. Congratulations on this interesting work.

DISTRIBUTION OF MATERIALS:

Again, congratulations on a very nice paper. I hope you found the review process to be constructive and are pleased with how the manuscript was handled editorially. We look forward to future exciting submissions from your lab.

Sincerely,
